# Clinical Significance of a Pain Scoring System for Deep Endometriosis by Pelvic Examination: Pain Score

**DOI:** 10.3390/diagnostics13101774

**Published:** 2023-05-17

**Authors:** Masao Ichikawa, Tatunori Shiraishi, Naofumi Okuda, Kimihiko Nakao, Yuka Shirai, Hanako Kaseki, Shigeo Akira, Masafumi Toyoshima, Yoshimitu Kuwabara, Shunji Suzuki

**Affiliations:** 1Department of Obstetrics and Gynecology, Nippon Medical School, 1-1-5 Sendagi, Bunkyo, Tokyo 113-8602, Japan; 2Department of Obstetrics and Gynecology, Nippon Medical School, Chibe Hokuso Hospital, 1715 Kamagari, Inzai 270-1694, Chiba, Japan; 3Meirikai Tokyo Yamato Hospital, 36-3 Honcho Itabashi, Tokyo 173-0001, Japan; 4Department of Obstetrics and Gynecology, Nippon Medical School, Musashikosugi Hospital, 1-383 Kosugicho, Nakahara, Kawasaki 211-8533, Kanagawa, Japan

**Keywords:** pain score, deep endometriosis, dysmenorrhea, dyspareunia, perimenstrual dyschezia, chronic pelvic pain, endometriotic nodule, obliteration of the pouch of Douglas, retroflexed uterus, adenomyosis

## Abstract

Endometriosis-associated pain is an essential factor in deciding surgical indications of endometriosis. However, there is no quantitative method to diagnose the intensity of local pain in endometriosis (especially deep endometriosis). This study aims to examine the clinical significance of the pain score, a preoperative diagnostic scoring system for endometriotic pain that can be performed only with pelvic examination, devised for the above purpose. The data from 131 patients from a previous study were included and evaluated using the pain score. This score measures the pain intensity in each of the seven areas of the uterus and its surroundings via a pelvic examination using a numeric rating scale (NRS) which contains 10 points. The maximum value was then defined as the max pain score. This study investigated the relationship between the pain score and clinical symptoms of endometriosis or endometriotic lesions related to deep endometriosis. The preoperative max pain score was 5.93 ± 2.6, which significantly decreased to 3.08 ± 2.0 postoperatively (*p* = 7.70 × 10^−20^). Regarding preoperative pain scores for each area, those of the uterine cervix, pouch of Douglas, and left and right uterosacral ligament areas were high (4.52, 4.04, 3.75, and 3.63, respectively). All scores decreased significantly after surgery (2.02, 1.88, 1.75, and 1.75, respectively). The correlations between the max pain score and dysmenorrhea, dyspareunia, perimenstrual dyschezia (pain with defecation), and chronic pelvic pain were 0.329, 0.453, 0.253, and 0.239, respectively, and were strongest with dyspareunia. Regarding the pain score of each area, the combination of the pain score of the pouch of Douglas area and the VAS score of dyspareunia showed the strongest correlation (0.379). The max pain score in the group with deep endometriosis (endometrial nodules) was 7.07 ± 2.4, which was significantly higher than the 4.97 ± 2.3 score obtained in the group without (*p* = 1.71 × 10^−6^). The pain score can indicate the intensity of endometriotic pain, especially dyspareunia. A local high value of this score could suggest the presence of deep endometriosis, depicted as endometriotic nodules at that site. Therefore, this method could help develop surgical strategies for deep endometriosis.

## 1. Introduction

Endometriosis is a disease that presents various pathological conditions, including endometrioma, endometriotic adhesions, and deep endometriosis (DE). Therefore, it is difficult to comprehensively evaluate all these pathologies simultaneously with a single diagnostic method [1,2,3]. To solve this problem, we developed a comprehensive diagnostic method for endometriosis: The Numerical Multi-scoring System for Endometriosis: NMS-E [4,5,6]. This method divides the pathology of endometriosis into four elements (endometrioma, adhesion, pain, DE-related lesions), evaluates the size, spread, and activity of the lesion using examination methods suitable for each pathology, and finally, it integrates them. It is a simple method that can be performed only with pelvic examination and transvaginal ultrasonography. The adhesion score to evaluate endometriotic adhesion, one of the main components of NMS-E, has already been reported [7]. This study refers to a preoperative diagnostic method for endometriotic pain called the pain score.

Endometriotic pain, such as dysmenorrhea and dyspareunia, is a crucial symptom when deciding on a surgery procedure, because it significantly reduces the patient’s quality of life. In order to perform the appropriate surgery, it is necessary to preoperatively identify the cause of the pain, investigate its size and spread, and then remove it effectively at the time of the surgery. Additionally, the activity of the lesion, which the pain intensity would reflect, is also essential information for specifying the appropriate resection area. However, to date, no diagnostic method exists to clarify this.

The Beecham classification [8] is a well-known diagnostic method that excels in detecting endometriotic pain. This method could be performed via a pelvic examination alone and captures early endometriotic lesions in the area of the pouch of Douglas (POD). Unfortunately, there is no specific form to systematically record its diagnostic result; therefore, this method cannot detect the extent of the lesion or the intensity of the pain. On the other hand, the widely used r-ASRM classification [9,10] is unsuitable for diagnosing deep lesions. Furthermore, since it is a classification method based on intraoperative findings, it does not function as a preoperative evaluation method. Additionally, the Enzian classification [11,12,13,14], which can complement r-ASRM with information on deep lesions, is not a preoperative diagnostic method. Therefore, a pain diagnosis method with a new perspective is necessary. The pain score evaluates the pain intensity in seven areas in the pelvis, centered on the uterus, on a 10-point numeric rating scale (NRS). Therefore, it is possible to identify the site that causes pain, its spread, and its activity.

The presence of DE has been emphasized as a pivotal factor that causes endometriotic pain [15]. The obliteration of the POD, a retroflexed uterus [16,17], endometriotic nodules [18], and adenomyosis [19] could be related to DE. The development of transvaginal ultrasonography and its diagnostic methods has made it easy to diagnose the above pathologies [20,21,22,23,24,25]. Therefore, a combination of pelvic examination and an imaging modality may help identify deep lesions.

This study aims to examine the pain score results and clarify the types of clinical symptoms and DE-related lesions that can be identified with this diagnostic method.

## 2. Materials and Methods

This diagnostic study was conducted using data from a previous prospective study [7] at Nippon Medical School Hospital. All patients included in previous studies were included in this study without exclusion. This study was approved by the ethics committee of Nippon Medical School. The pain score of this evaluation method was performed by one surgeon (M.I.). A pain score assessment was performed on admission or immediately prior to surgery. Simultaneously, transvaginal ultrasound was also routinely performed to detect endometrial lesions. The transvaginal ultrasound device used was a Voluson E8 (GE Healthcare Japan). Depending on the case, magnetic resonance imaging (MRI) was performed.

### 2.1. Pain Score and Max Pain Score

The pain score is an examination method to detect endometriosis-associated pain by pelvic examination alone. The uterine cervix and surrounding regions are divided into seven areas, and the intensity of pain in each area is evaluated on a scale of 10 using the NRS. The results are listed on the pain score map (Figure 1). The pain score map consists of squares in a 3 × 3 grid. This map records the pain intensity from the pelvic examination at anatomically matched locations. The seven areas of the pelvis are the anterior vagina wall (A), right and left adnexa (rO and lO), uterine cervix (Cx), right and left uterosacral ligaments (rU and lU), and POD (D). The stimulation method during the pelvic examination entails first lifting the uterine cervix upward with the index finger in the vagina. The NRS value of pain caused by the stimulation is the pain score in the uterine cervix area. Next, push the POD backward. Then, flick the right USL and left USL downwards or upwards with the index finger to stimulate those areas. Finally, evaluate the pain scores caused by each stimulus via compression of the right or left adnexal area by bimanual examination or anterior vaginal wall compression with the index finger in the vagina. The maximum intensity is defined as the max pain score among the seven areas. In the presented case, there is severe pain in the left and right uterosacral ligament areas, but the max pain score is 8 points in the right uterosacral ligament area.

### 2.2. Evaluation of Endometriotic Lesions

Retroflexed-uterus cases are those in which the angle of the cervix and the uterus axis at the rear of the uterus is less than 180 degrees on MRI or transvaginal ultrasonography [26]. On transvaginal ultrasonography, endometrial nodules were depicted as hypoechoic areas around the uterus [27,28]. Lesions larger than 1 cm are diagnosed as endometriotic nodules. In addition, a palpable b-b shot nodule or a larger nodule by pelvic examination [29] is also considered an endometriotic nodule. Adenomyosis is diagnosed when there are hyperechoic and hypoechoic luminous speckled patterns on transvaginal ultrasonography.

### 2.3. Statistics

The correlation coefficient was used to calculate the correlation between the pain score by NRS and the visual analog scale (VAS) value by interview using the CORREL function in Excel software, version 2021. Student’s *t*-test (two-tailed test) was used to assess significant differences in pain between the groups with and without endometriotic lesions, using the TTEST function in Excel software, version 2021.

## 3. Results

### 3.1. Pain Score Results

The total mean age of the patients was 35.6 years (ranging from 23 to 51 years). There were 61 patients with unilateral endometriomas and 68 with bilateral endometriomas. All these lesions were treated mainly by cystectomy (see reference [7] for details). Table 1 shows the results of the mean pain scores before and after surgery in 131 patients. The mean preoperative max pain score was 5.93 ± 2.6. When compared by area, the highest average pain score was 4.52 in the uterine cervix area, followed by 4.04 in the POD area, 3.75 in the left uterosacral ligament, and 3.63 in the right uterosacral ligament area. In contrast, the anterior vaginal wall and the left and right adnexal areas had low pain scores. Postoperatively, the mean max pain score dropped to 3.08 ± 2.0, with a significant difference (7.70 × 10^−20^). In all areas except the right adnexal area, where the average pain score was initially low, pain scores decreased with high significance, achieving values of almost two or less.

At surgery, 94 of 131 patients (71.8%) had partial or complete obliteration of POD. All but nine of these cases underwent a complete resolution of POD obliteration ± excision of DE. Therefore, it was shown that the pain score could quantitatively measure the pain reduction effect of surgical lesion removal for each area.

Appendix A presents the change in group distribution based on the severity classification of the max pain score before and after surgery. The percentage of patients in the severe pain group (max pain score of 8–10 points) decreased from 29.8% (39) to 3.1% (4), while pain-free patients (max pain score of 0 points) increased from 1 to 10. Meanwhile, Appendix A shows each patient’s postoperative max pain score, arranged in ascending order, overlaid with the preoperative max pain score. Even in cases where the preoperative max pain score was in the severe range, there were many cases in which the pain decreased significantly after the operation. Approximately 55.7% of patients improved their max postoperative pain score by 50% or more, and 70.2% improved by 30% or more. Overall, 83.2% (109) of patients showed improvement in their max pain score one month after surgery, while 6.1% (8) remained unchanged, and 10.7% (14) experienced worsening (Appendix A). Among the 14 patients in the worsened group, we followed up with 12 cases after 6 or 12 months. Five showed improvements, three remained the same, and four worsened compared to their pre-surgery status.

### 3.2. Association between Pain Score and Clinical Symptoms

Next, we then examined the type of clinical symptoms the pain score reflects by investigating the correlation between the pain score and the dysmenorrhea, dyspareunia, perimenstrual dyschezia (pain with defecation), and chronic pelvic pain that are often observed in endometriosis (Table 2). The mean preoperative VAS values by interview for dysmenorrhea, dyspareunia, perimenstrual dyschezia, and chronic pelvic pain were 6.63 (*n* = 119), 3.52 (*n* = 110), 2.94 (*n* = 119), and 2.07 (*n* = 119), respectively. The mean preoperative max pain score was 5.99, *n* = 119 (6.25, *n* = 110 in dyspareunia), close to the VAS value of dysmenorrhea. However, the max pain score was most strongly correlated with dyspareunia (r = 0.453). Regarding local pain scores, the combination with the strongest correlation was the pain score in area D and dyspareunia (r = 0.379). In contrast, there was little or very low association between the pain score and perimenstrual dyschezia or chronic pelvic pain (−0.027 to 0.221).

Figure 2 shows a scatterplot of the correlation between the max pain score and dyspareunia, as well as between the max pain score and dysmenorrhea. There are more points on the *Y*-axis in the scatterplot with dyspareunia than with dysmenorrhea. This suggests that the max pain score may have a higher sensitivity than the dyspareunia evaluation by VAS interview.

### 3.3. Association between Pain Score and Endometriotic Lesions

Thirdly, we investigated the type of endometriotic lesions with which the pain score is strongly associated. The types of lesions examined were POD obliteration (including partial and complete obliteration), retroflexed uterus, DE (endometriotic nodules), and adenomyosis. The comorbidity rates for each type of lesion were 71.7% (94 cases), 48.9% (64 cases), 45.8% (60 cases), and 19.1% (25 cases), respectively. Table 3 shows the results of comparing pain scores between the groups with and without lesions. Comparing the max pain scores in the presence and absence of the four types of lesions, the group with DE showed a significantly higher score than the group without DE (7.07 ± 2.4 vs. 4.97 ± 2.3 *p* = 1.71 × 10^−6^). There was no significant difference in the presence or absence of the other types of lesions.

A comparison of the pain scores in each area with and without the four types of lesions showed that the POD obliteration group showed significantly higher pain scores in the D and rU areas (4.46, *p* = 0.006 and 3.95, *p* = 0.03). Regarding the retroflexed uterus, a significantly higher pain score was shown in the D area. DE showed significantly higher pain scores in the rU, D, and lU areas. DE in this study was defined as DE+ when there were endometrial nodules which were detected as lesions showing low echogenicity on transvaginal ultrasonography (Figure 3) or palpated by pelvic examination. No significant difference was observed for adenomyosis.

Taken together, the max pain score was shown to be higher in the presence of DE. It was also shown that the presence of DE, POD obliteration, and retroflexion of the uterus were closely associated with pain around the POD area.

### 3.4. Association between Surgical Resection of DE-Related Lesions and Pain Score

Finally, we examined the relationship between surgical resection of DE-related lesions and changes in pain score. Table 4 shows preoperative (upper low) and postoperative (middle low) mean pain scores for each area in two groups: those with partial or complete obliteration of the POD, which was resolved entirely by surgery, and those without lesions in the POD. The group that underwent surgical resolution of obliterated POD had the largest significant difference in pain score compared to the group without lesions at the POD, in region D (4.72 vs. 2.97, *p* = 0.001) before surgery. Although the pain score decreased significantly in the same area in both groups after surgery, the significant difference between the two groups disappeared (1.91 vs. 1.79, *p* = 0.69) postoperatively.

We also investigated the effect of deep lesion resections in the USL area on pain score (Table 5). Pain score comparisons were made among patients with normal bilateral USL regions, ipsilateral resection of unilateral USL with suspected DE, and bilateral resections of USLs with suspected DE. In the unilateral resection group, the pain score on the affected side tended to be higher (Group 2 with affected right USL: 4.82 vs. 3.00, Group 3 with affected left USL: 2.50 vs. 4.00), but the difference was not significant. We divided the USLs of all patients into healthy and affected samples and found that the preoperative pain scores on the affected USLs were significantly higher (4.39 vs. 2.38 *p* = 1.41 × 10^−8^). However, the difference between the two groups disappeared after surgery (1.71 vs. 1.72 *p* = 0.97). Of the 171 removed USL specimens, 167 were pathologically diagnosed (excluding crushed specimens), and 143 samples (85.6%) had confirmed endometriotic tissue.

## 4. Discussion

The present study found that of the seven areas, the uterine cervix had the highest pain score (4.52), followed by POD (4.04) and then the left and right uterosacral ligament areas (3.75 and 3.63, respectively). These areas form a posteriorly-radiating fan-shaped area originating at the cervix. In contrast, the pain scores in the anterior, left, and right adnexal areas were found to be relatively low. We hereby consider the meaning of the pain score in each area. First, why is the pain score of the cervical area the highest? The pain at this site is measured by the intensity of movement pain caused by cervical elevation. Since uterine cervix elevation inevitably stimulates lesions in the posterior areas, the pain score in the uterine cervix area can be regarded as a value that summarizes pain in the posterior area. That may be why this area has the highest pain score value. In contrast, pain scores in the POD and left and right uterosacral ligament areas are thought to reflect more localized lesion pain. Therefore, a high pain score in these areas can be an essential index for determining lesion resection in that area.

On the other hand, the left and right adnexal areas had unexpectedly low pain scores. In fact, of the 61 cases with unilateral endometrioma, the mean pain scores in the adnexal area on the side with and without cysts were 2.02 and 2.05, respectively, with no significant difference between the two groups (*p* = 0.94). Therefore, the presence of endometrioma alone is not thought to cause pain. Since area A of the anterior wall has relatively few lesions, the pain score value in this area may play a negative control role in this method.

The Numeric Rating Scale (NRS) is commonly used to interpret pain score values. According to the National Comprehensive Cancer Network (NCCN) guidelines, pain severity is classified as follows: 0 is considered normal, 1–3 indicates mild pain, 4–7 indicates moderate pain, and 8–10 indicates severe pain in NRS [30]. In cancer pain, a score of 1–3 indicates pain intensity that can be treated with NSAIDs, while a score of 4 or higher indicates pain intensity that may require opioids. Therefore, a local pain score of 4 points or more is considered an index of pain that should be addressed through surgery or drugs other than NSAIDs, even in the case of endometriosis pain. In this study, the preoperative mean pain score in the group where the USL region was surgically resected was 4.39 (Table 5), which indicates significant pain that required treatment beyond NSAIDs. Additionally, 85.6% of the specimens were pathologically confirmed to have endometriosis, supporting the judgment for USL resection in this study.

The max pain score was most strongly correlated with dyspareunia among the four major clinical symptoms of endometriosis. It was also found that the region closely related to dyspareunia was the POD. These results suggest that lesions around the POD area may cause dyspareunia and the pain score in the POD is effective as an index for evaluating dyspareunia. However, if there is another painful area besides the POD, that area determines the intensity of dyspareunia. Therefore, recognizing where the most painful point is on the pain score map at the time of surgery and reliably excising the lesion at that point is key to improving postoperative dyspareunia. That is the strength of this method, which was not obtained in the dyspareunia evaluation by VAS interview.

In recent years, less sex has become a social problem. According to a report from the UK, 29.3% (2012) of women say they do not have sex at least once a month [31]. The percentage of sexless women in Japan is as high as 49.5% [32]. Therefore, there is a considerable number of cases in which it is impossible to get information about dyspareunia by interviewing patients with endometriosis. Even if it could be attained, the question remains about how accurately the scores reflect actual dyspareunia. This may be why dyspareunia by VAS interview in this study was 3.52, which was lower than the max pain score of 5.99, and why there were many points on the *X*-axis in the distribution of the correlation diagram (Figure 2b). Therefore, it is possible to use this method to solve the problem of sexlessness and reduce the possibility of overlooking lesions that cause potential dyspareunia.

It was somewhat unexpected that the correlation between the pain score and perimenstrual dyschezia was very low in the results of this study. The rectum frequently adheres to the posterior part of the lower part of the uterus. We therefore assumed that this adhesion might be the cause of painful defecation. Postoperative perimenstrual dyschezia and chronic pelvic pain decreased significantly (data not shown). It is therefore necessary to examine what causes perimenstrual dyschezia.

Next, we discuss the relationship between max pain score and deep lesions. The pain score was compared with lesions associated with pain, such as POD obliteration, retroflexed uterus, DE identified by transvaginal ultrasonography or pelvic examination as endometrial nodules, and adenomyosis. As a result, we found that DE (endometriotic nodules) were the most critical factors that caused high pain scores. This finding is significant because it allows us to visually and anatomically identify the triggering sites of endometriosis-associated pain. It allows us to define actual targets for resection during surgery. Moreover, the pain score enables the detection of the activity of the lesion by measuring pain. The intensity of its activity is essential in deciding whether to resect the lesion, especially in peri-menopausal patients. Furthermore, conditions such as POD obliteration and a retroflexed uterus were also shown to be associated with pain in the POD area. These facts also indicate that in order to improve dyspareunia, it is vital to remove these pathological conditions appropriately. In this study, it was found that the pain score in the areas where the obliterated POD was resolved or where the USL with deep lesions was excised significantly decreased after surgery. This indicates the importance of reliably removing such deep lesions.

Finally, we summarize how to use this method in surgery and clinics. When a deep lesion is observed during surgery, it is sometimes difficult to judge the intensity of its activity only from the endoscopic image. VAS information from interviews cannot determine the extent of resection because it is impossible to identify areas that provoke pain. On the other hand, the pain score divides the pelvis into seven regions and can provide pain information in those regions. In particular, pain in the USL region or the POD region suggests local involvement. Furthermore, the present study also showed that these lesions were associated with dyspareunia. These facts indicate that the pain score information can indicate lesion activity, which is one of the criteria for deciding whether to resect the lesion. For example, when deciding whether to resect the USL bilaterally or only unilaterally, it is possible to make a more accurate decision by integrating endoscopic images and pain score information. There were also nine cases of incomplete surgery in this study. Five of these had complete obliteration of POD, but the pain score in that area was as low as 3 points or less, and menopause was approaching. Therefore, we deliberately did not perform complete resolution of the lesions in these cases. In this way, optimizing the surgical content according to the conditions of individual cases can reduce the risk of complications due to excessive surgery.

This pain score could be useful for a wide range of conditions of endometriosis, from severe to mild. Deep lesions are present in most severe cases. The pain score serves as an index for determining its localization, spread, and activity, and serves as a criterion for determining the excision range. It is also effective in mild cases. Small endometriotic lesions are difficult to diagnose with transvaginal ultrasound, especially in cases that do not have endometriomas. Even in such cases, the pain score can identify localized pain and is an effective indicator for surgical decisions. As with the Beecham classification [8], pain assessments by pelvic examination are important to identify early lesions of endometriosis.

In addition, since this method is non-invasive and easy to use, and the postoperative course can be followed over time, it could help monitor for recurrence and determine the necessity of drug administration.

As mentioned at the beginning, the pain score is one of the main components of the Numerical Multi-scoring System for Endometriosis: NMS-E [4,5,6,7], which comprehensively diagnoses endometriosis using pelvic examination and transvaginal ultrasonography. Therefore, in practice, the overall picture of endometriosis is diagnosed not only from the findings of the pelvic examination but also from the findings of the transvaginal ultrasonography.

The limitation of this study is that there was only one examiner. As a result, we obtained consistent data, but the possibility of bias was fully considered. Therefore, to prove that this method is universal, it is necessary to check the reproducibility of this data among many examiners and facilities and confirm its effectiveness. We are also planning an initiative to measure inter-rater reliability at our institution. Another problem was the small number of study cases. Since endometriosis is a disease showing various pathologies, many confounding factors exist. Therefore, in the future, it is necessary to increase the number of cases and make adjustments through matching and stratification.

## 5. Conclusions

The pain score can be a new indicator of the intensity of dyspareunia. A local high value of this test suggests the presence of DE, such as an endometrial nodule, at that site. Therefore, this method serves as a valuable index for localizing lesions that cause dyspareunia and for determining the lesion’s excision range in surgical strategies for endometriosis.

## Figures and Tables

**Figure 1 diagnostics-13-01774-f001:**
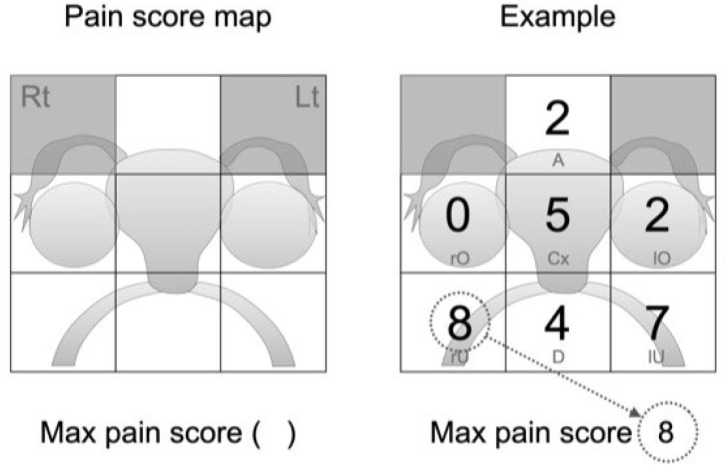
Pain score map and Example.

**Figure 2 diagnostics-13-01774-f002:**
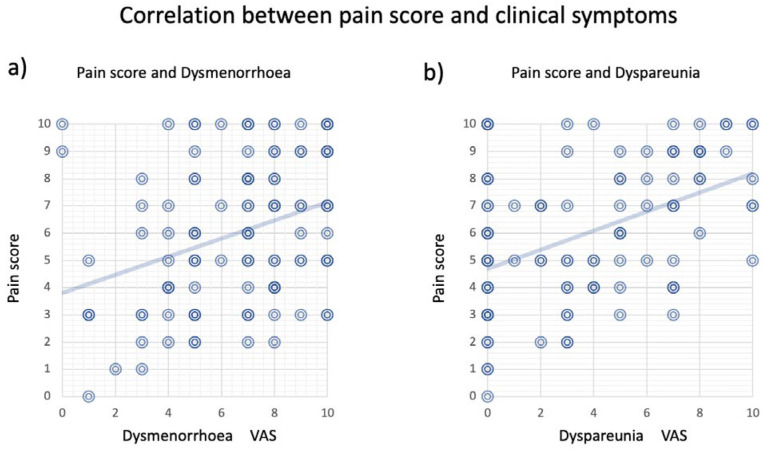
Correlation between max pain score and clinical symptoms. (**a**) Correlation between max pain score and intensity of dysmenorrhea by VAS. (**b**) Correlation between max pain score and dyspareunia by VAS. The double circles in this scatterplot represent individual cases, and the intensity of the color indicates the overlap of cases. The diagonal line represents the regression line.

**Figure 3 diagnostics-13-01774-f003:**
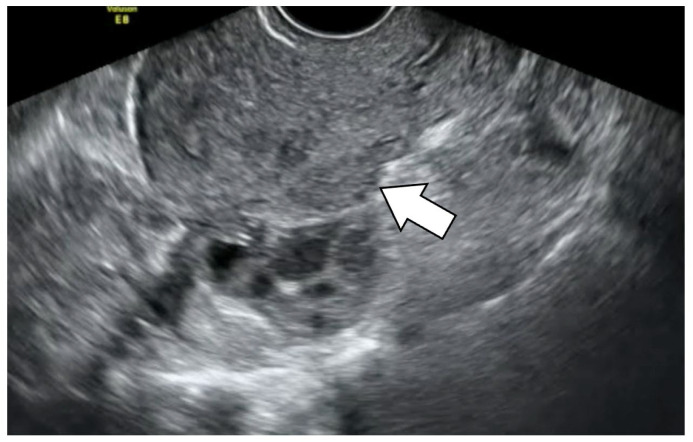
Endometriotic nodule on transvaginal ultrasonography. The white arrow points to an endometriotic nodule about 2 × 1 cm in size, which is slightly less bright and is present on the posterior surface of the uterus.

**Table 1 diagnostics-13-01774-t001:** Mean pain score values for each pelvic area during pelvic examination before and after surgery.

	Pain Score							
	Max	A	rO	Cx	lO	rU	D	lU
Preoperative Mean pain score (*n* = 131)	5.93	2.07	1.92	4.52	2.18	3.63	4.04	3.75
Postoperative Mean pain score (*n* = 131)	3.08	1.40	1.53	2.02	1.28	1.75	1.88	1.75
*p* value	7.70 × 10^−20^	0.0028	0.13	1.78 × 10^−16^	0.0004	2.59 × 10^−10^	3.27 × 10^−12^	3.45 × 10^−11^

Abbreviation Max: Max pain score, A: anterior vaginal wall area, rO: right adnexal area, Cx: uterine cervix area, lO: left adnexal area, rU: right uterosacral ligament area, D: POD area, lU: left uterosacral ligament area.

**Table 2 diagnostics-13-01774-t002:** Association between pain score and clinical symptoms.

		Pain Score						
	Max	A	rO	Cx	lO	rU	D	lU
Dysmenorrhoea (*n* = 119)	0.329 *	0.237	0.272	0.215	0.145	0.267	0.206	0.307
Dyspareunia (*n* = 110)	0.453	0.234	0.183	0.375	0.095	0.350	0.379	0.190
Dyschezia (*n* = 119)	0.253	0.141	0.216	0.050	−0.027	0.219	0.148	0.221
Chronic pelvic pain (*n* = 119)	0.239	0.076	0.216	0.222	0.025	0.118	0.150	0.192

Abbreviation Max: Max pain score, A: anterior vaginal wall area, rO: right adnexal area, Cx: uterine cervix area, lO: left adnexal area, rU: right uterosacral ligament area, D: POD area, lU: left uterosacral ligament area. * Correlation coefficient: r.

**Table 3 diagnostics-13-01774-t003:** Association between pain score and endometriotic lesions.

		Pain Score							
		Max	A	rO	Cx	lO	rU	D	lU
POD	Normal (*n* = 37)	5.24	1.86	2.03	4.19	2.11	2.81	2.97	3.41
	Complete or Partial obliteration of POD (*n* = 94)	6.20	2.15	1.87	4.65	2.21	3.95	4.46	3.88
	*p* value	0.06	0.47	0.72	0.38	0.81	0.03	0.006	0.37
Uterus	Retroflexion *: <180 (*n* = 64)	6.30	2.14	1.98	4.52	2.23	4.00	4.88	3.97
	Anteflexion: ≥180 (*n* = 67)	5.58	2.00	1.85	4.52	2.13	3.27	3.24	3.54
	P value	0.12	0.69	0.73	0.99	0.80	0.12	0.0007	0.36
	DE ** + (*n* = 60)	7.07	2.28	2.08	5.32	2.35	4.62	4.85	4.40
	DE − (*n* = 71)	4.97	1.89	1.77	3.85	2.04	2.79	3.35	3.20
	*p*-value	1.71 × 10^−6^	0.27	0.43	0.00	0.44	7.28 × 10^−5^	0.002	0.011
	Adenomyosis *** + (*n* = 25)	6.40	1.96	1.88	5.04	2.12	4.12	4.24	4.52
	Adenomyosis − (*n* = 106)	5.82	2.09	1.92	4.40	2.20	3.51	3.99	3.57
	*p*-value	0.32	0.77	0.93	0.29	0.88	0.31	0.69	0.11

Abbreviation POD: Pouch of Douglas, DE: Deep endometriosis, Max: Max pain score, A: anterior vaginal wall area, rO: right adnexal area, Cx: uterine cervix area, lO: left adnexal area, rU: right uterosacral ligament area, D: POD area, lU: left uterosacral ligament area. * Retroflexion refers to cases where the angle between the cervical axis and the uterine corpus axis in the posterior uterus is less than 180°. ** DIE refers to endometrial nodules detectable by transvaginal ultrasonography. *** Adenomyosis refers to endometrial nodules detectable by transvaginal ultrasonography or MRI.

**Table 4 diagnostics-13-01774-t004:** Association between obliterated POD and pain scores in endometriosis patients undergoing surgery.

	Pain Score							
	Max	A	rO	Cx	lO	rU	D	lU
1. Preoperative Mean pain score in cases with normal POD * (*n* = 37)	5.24	1.86	2.03	4.19	2.11	2.81	2.97	3.41
2. Preoperative Mean pain score in cases with obliterated POD ** (*n* = 85)	6.52	2.20	1.94	4.91	2.25	4.13	4.72	4.06
*p* value	0.0098	0.40	0.85	0.18	0.76	0.01	0.001	0.23
1′. Postoperative Mean pain score in cases with normal POD *** (*n* = 37)	3.30	1.51	1.59	2.46	1.35	1.89	1.76	2.11
2′. Postoperative Mean pain score in cases with obliterated POD *** (*n* = 85)	3.02	1.31	1.52	1.82	1.22	1.67	1.91	1.52
*p* value	0.49	0.5	0.86	0.08	0.71	0.55	0.69	0.11
1 vs. 1’ (*n* = 37) ^†^ *p*-value	0.0007	0.36	0.41	0.001	0.09	0.10	0.03	0.02
2 vs. 2’ (*n* = 85) ^†^ *p*-value	9.66 × 10^−20^	0.002	0.18	2.72 × 10^−15^	0.002	4.83 × 10^−11^	6.03 × 10^−13^	2.43 × 10^−11^

Abbreviation Max: Max pain score, A: anterior vaginal wall area, rO: right adnexal area, Cx: uterine cervix area, lO: left adnexal area, rU: right uterosacral ligament area, D: POD area, lU: left uterosacral ligament area. * The subjects were 37 cases in which no obliteration was observed in thePOD at the time of surgery. ** The subjects cases were 85 cases in which partial or complete obliteration of the POD was observed and the resolution of it was completed at the time of surgery. *** Mean Pain score at 1 month after surgery in the same group as 1 and 2. † Examination of the significant difference between the preoperative and postoperative pain scores of group 1 or 2. To compare the preoperative and postoperative pain scores of groups, a two-tailed paired *t*-test was conducted.

**Table 5 diagnostics-13-01774-t005:** Pain score changes in main areas following USL resection.

	Pain Score							
	Max	Cx	rU	D	lU	Healthy USLs	Affected USLs	*p* value
1. Preoperative Mean pain score in cases without USL resections (*n* = 25)	3.92	3.08	2.12	2.24	2.28	-	-	0.76 *
2. Preoperative Mean pain score in cases with right USL resections alone (*n* = 11)	5.55	4.36	4.82	4.09	3.00	-	-	0.18 *
3. Preoperative Mean pain score in cases with left USL resections alone (*n* = 20)	5.75	4.30	2.50	3.95	4.00	-	-	0.06*
4. Preoperative Mean pain score in cases with bilateral USL resections (*n* = 70)	6.94	5.27	4.40	4.83	4.43	-	-	0.95 *
Preoperative Mean Pain Score for Individual USLs w/o or w/Lesions (*n* = 81 vs. 171) **	-	-	-	-	-	2.38	4.39	1.41 × 10^−8^
1′. Postoperative Mean pain score in cases without USL resections (*n* = 25)	2.72	2.04	1.52	1.52	1.96	-	-	0.34 *
2′. Postoperative Mean pain score in cases with right USL resections alone (*n* = 11)	3.18	2.18	1.55	2.00	1.55	-	-	1.00 *
3′. Postoperative Mean pain score in cases with left USL resections alone (*n* = 20)	2.90	2.20	1.75	1.65	1.60	-	-	0.81 *
4′. Postoperative Mean pain score in cases with bilateral USL resections (*n* = 70)	3.24	1.93	1.81	2.03	1.66	-	-	0.60 *
Postoperative Mean Pain Score for Individual USLs w/o or w/Lesions (*n* = 81 vs. 171) **	-	-	-	-	-	1.72	1.71	0.97

Abbreviation USL: uterosacral ligament, Max: Max pain score, Cx: uterine cervix area, rU: right uterosacral ligament area, D: POD area, lU: left uterosacral ligament area. * This study evaluated the maximum pain score in 126 patients, excluding five patients who underwent incomplete surgery. This P value indicates a significant difference in comparing pain scores in the rU and lU regions. ** In this study, one endometriosis patient was considered to have two samples of the right and left USL regions. If the bilateral USLs were normal, the case was rated as having two healthy samples. Cases with only unilateral lesions were rated as having one healthy sample and one sample with lesions. Finally, cases with bilateral lesions were rated as having two samples with lesions. In this way, 252 samples from 126 patients, excluding five patients who underwent incomplete surgery, were divided into a healthy group of 81 patients and a lesioned group of 171 patients and compared.

## Data Availability

Due to ethical considerations and restrictions, we regretfully cannot share the research data underlying the reported results. The data contains sensitive information that must be protected under privacy and ethical guidelines. We apologize for any inconvenience this may cause. For further information or inquiries regarding the study, please contact [masai@nms.ac.jp].

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
