# Peer review of "Clinical Significance of a Pain Scoring System for Deep Endometriosis by Pelvic Examination: Pain Score"

_diagnostics, 2023, doi:10.3390/diagnostics13101774_

Round 1
Reviewer 1 Report
The authors present an endometriosis associated pain score and investigaste its clinical significance. Endometriosis is a very common medical entity that may cause a variety of symptoms and may have a significant impact on the patient's daily life. i have a few comments, that i believe, the authors should include in their manuscript.
Comment 1
Did any of the patients mention a postoperative pain score 0? in which grou of patients did they belong?
Comment 2
Which patient group reported greater relief of symptoms after surgery?
Comment 3
The authors mention "We herein refer to the interpretation of pain score values in general NRS. A score of 0 is considered normal, 1–3 mild pain, 4–6 moderate pain, and 7–10 severe pain. A score of 4 or more requires some treatment. Judging from this criterion, a lesion in a area with a high pain score of 4 points or more means it needs surgical treatment."
- Please put a reference.
- Why should the treatment be surgical?
-Why 4 is a cutoff score for surgical treatment according to your statistical analysis?
Comment 4
Were there any patients who intraoperatively had more endometriotic lessions in other areas than the ones that they mentioned the maximum pain preoperatively?
Comment 5
Do the authors believe that this pain score is more usefull and reliable in cases of deep infiltrating endometriosis or in any case of endometriosis?
Author Response
Dear Reviewer 1
We sincerely thank you for your valuable comments and suggestions on our manuscript.
We changed it according to your comments like the following.
Comment 1
Did any of the patients mention a postoperative pain score of 0? In which group of patients did they belong?
→ Yes, I had ten cases. I added the information on the text (page.4, line 214) and supplementary data: Fig. S1 shows the individual change.
Comment 2
Which patient group reported greater relief of symptoms after surgery?
→ Fig. S1 and two also show the above information regarding the Max pain score.
And I added two tables (4 and 5). Those data show resolution of obliterated POD or resection of affected USLs can affect which place's pain was relieved.
Comment 3
The authors mention, "We herein refer to the interpretation of pain score values in general NRS. .... with a high pain score of 4 points or more means it needs surgical treatment."
- Please put a reference.
→ I did. Ref.ï¼»30ï¼½
- Why should the treatment be surgical?
→ We introduced National Comprehensive Cancer Network (NCCN) guidelines in the discussion on page 8, line 406. In that, 4 or more pain could need stronger treatment over NSAIDs in cancer treatment. But, surgical treatment is not the only option. Other drugs like Ginodiest could be another option depending on the type of patient.
-Why is 4 a cutoff score for surgical treatment according to your statistical analysis?
I added Table 5 to answer this. My data showed that the patient who removed the affected USL had a higher score in those areas: 4.39. Score 4 could be the indicator to change how to treat to some extent.
Comment 4
Were there any patients who intraoperatively had more endometriotic lesions in other areas than the ones that they mentioned the maximum pain preoperatively?
→ Yes. I have added the information about this. 4-page line:224.
Comment 5
Do the authors believe that this pain score is more useful and reliable in cases of deep infiltrating endometriosis or, in any case of endometriosis?
Yes. This approach could be helpful in a wide range of conditions of endometriosis, from severe to mild. I mentioned my thought on page 10, line 525.
Thank you for your comments and for taking your valuable time for me.
Once again, we would like to thank you for your time and effort in reviewing our manuscript.
Reviewer 2 Report
Thank you for inviting me to review this manuscript.
The paper itself is well written.
The authors undertaken a rigorous piece of data collection and have analyze information accurately.
The article is very interesting. Material and methods are correct. The conclusions are clear and consistent logically.
Moreover the authors conducted a thorough literature revew.
It was a pleasure to read this manuscript.
Author Response
Dear Reviewer
We sincerely thank you for your valuable comments on our manuscript.
We changed it to improve the content according to comments from other reviewers. So attached the latest version of it.
And this time, I checked the English of my manuscript by using the MDPI editing service.
Once again, we would like to thank you for your time and effort in reviewing our manuscript.
Reviewer 3 Report
This paper is about a pain scoring system for deep endometriosis in pelvic examinations.
The paper describes new findings, including that cervical pain scores are high for pain and that dyspareunia correlates most strongly with pain scores.
However, the main drawbacks of this paper, as the authors note in their limitation, are the relatively small number of cases, the subjective nature of the examination method, which is mainly based on examination findings, and the lack of diagnostic accuracy because only one examiner was used.
I understand that Diagnostics is a journal that aims to increase the accuracy of diagnosis and that the conventional classification does not correspond to pain, but it is a little unclear what benefit the reader will get from using this new diagnostic method.
To improve this paper, improvements would include the following
Can similar diagnoses be made by different examiners and are the results reproducible? We believe that the validation needs to be verified.
L110 2.1 -> 2.2 and needs to be revised.
Shouldn't the content of L167 be included in the discussion?
Author Response
Dear Reviewer
We sincerely thank you for your valuable comments and suggestions on our manuscript.
We changed it according to the reviewer's comments.
We attached a revised ver. of it here. And also, this time, We checked our manuscript through the MDPI editing.
Answers to comments.
However, the main drawbacks of this paper, as the authors note in their limitation, are the relatively small number of cases, the subjective nature of the examination method, which is mainly based on examination findings, and the lack of diagnostic accuracy because only one examiner was used.
→ We agree with your suggestion. We are also planning an initiative to measure inter-rater reliability at our institution. We mentioned that in the discussion: on page 10, line 547. So, hopefully, we can provide those data next time.
I understand that Diagnostics is a journal that aims to increase the accuracy of diagnosis and that the conventional classification does not correspond to pain, but it is a little unclear what benefit the reader will get from using this new diagnostic method.
→We added the new data (Table. 5), stating the reason for the need for localized or personalized treatment based on the pain score. And we added a new paragraph mentioning how to use this method in the discussion on Page 9, line 491.
Can similar diagnoses be made by different examiners, and are the results reproducible? We believe that the validation needs to be verified.
→ Sorry again. We are planning an initiative to measure inter-rater reliability at our institution now. Before this method, we published the adhesion score for endometriosis ï¼»ref. 7ï¼½. And about it, we reported no difference in the evaluation of adhesions by transvaginal ultrasound among three examiners (the intraclass correlation coefficient among examiners is 0.87). ï¼»ref. 5ï¼½
L110 2.1 -> 2.2 and needs to be revised.
→Thanks. We did.
Shouldn't the content of L167 be included in the discussion?
→ Thanks. We did.
Thank you for your comments and for taking your valuable time for me.
We would like to thank you for your time and effort in reviewing our manuscript.
Round 2
Reviewer 3 Report
The part I was pointing out has been corrected.
It seems to be acceptable.